# Taxanes in the Treatment of Head and Neck Squamous Cell Carcinoma

**DOI:** 10.3390/biomedicines11112887

**Published:** 2023-10-25

**Authors:** Ching-Yun Hsieh, Ching-Chan Lin, Wei-Chao Chang

**Affiliations:** 1Division of Hematology and Oncology, Department of Internal Medicine, China Medical University Hospital, China Medical University, Taichung 40402, Taiwan; d7995@mail.cmuh.org.tw; 2Center for Molecular Medicine, China Medical University Hospital, China Medical University, Taichung 40402, Taiwan

**Keywords:** head and neck squamous cell carcinoma (HNSCC), taxanes, docetaxel (DTX), resistance

## Abstract

Taxanes, particularly docetaxel (DTX), has been widely used for combination therapy of head and neck squamous cell carcinoma (HNSCC). For locally advanced unresectable HNSCC, DTX combined with cisplatin and 5-fluorouracil as a revolutionary treatment revealed an advantage in the improvement of patient outcome. In addition, DTX plus immune check inhibitors (ICIs) showed low toxicity and an increased response of patients with recurrent or metastatic HNSCC (R/M HNSCC). Accumulated data indicate that taxanes not only function as antimitotics but also impair diverse oncogenic signalings, including angiogenesis, inflammatory response, ROS production, and apoptosis induction. However, despite an initial response, the development of resistance remains a major obstacle to treatment response. Taxane resistance could result from intrinsic mechanisms, such as enhanced DNA/RNA damage repair, increased drug efflux, and apoptosis inhibition, and extrinsic effects, such as angiogenesis and interactions between tumor cells and immune cells. This review provides an overview of taxanes therapy applied in different stages of HNSCC and describe the mechanisms of taxane resistance in HNSCC. Through a detailed understanding, the mechanisms of resistance may help in developing the potential therapeutic methods and the effective combination strategies to overcome drug resistance.

## 1. Introduction

Head and neck squamous cell carcinoma (HNSCC) is the sixth most common type of cancer worldwide, with an annual incidence of more than 890,000 cases [1]. HNSCC develops in the outer layer of the skin and in the mucosal epithelium of the mouth, nose, and throat and impairs important physiological functions such as breathing, swallowing, and vocalization depending on its location. Common risk factors for HNSCC are associated with human papillomavirus infection and dietary habits such as betel nut chewing, heavy cigarette smoking, and excessive alcohol consumption [2]. Due to the lack of appropriate screening biomarkers and unobvious or nonspecific symptoms in early-stage HNSCC, patients are often diagnosed in an advanced state with a poor prognosis and a 5-year overall survival (OS) rate of <50% [3,4,5]. Multidisciplinary treatments, including surgery, targeted therapy, radiotherapy, chemotherapy, and immunotherapy, have been considered for patients with HNSCC according to the stage of disease, anatomical site, and surgical accessibility [5]. Among these therapeutic methods, chemotherapeutic drugs remain the major regimens for systemic treatment. Currently, a combination of docetaxel (DTX) with cisplatin plus 5-fluorouracil (TPF) has been the most popular therapeutic strategy for HNSCC [6]. Clinical verification revealed that complete response to TPF treatment was significantly associated with longer progression-free survival (PFS) and OS in patients with HNSCC [7,8].

Taxanes with a core structure of taxadiene are a class of diterpenes originally identified from plants of the genus *Taxus* (yews). They represent one of the most interesting categories of antineoplastic drugs; paclitaxel (PTX) and the synthetic derivatives DTX and cabazitaxel (CTX) are FDA-approved taxanes that are currently used in the treatment of various cancers (Figure 1) [9]. In addition, novel taxanes, particularly oral taxanes, attract the most attention and are under intensive investigation in preclinical and clinical settings. Oral taxanes show the advantages of increased patient convenience, improved bioavailability, decreased toxicity profile, and maintaining activity in tumors expressing the multidrug-resistance gene [10,11]. The newly developed oral taxanes tesetaxel (DJ-927) and larotaxel (XRP9881) have been evaluated by phase III clinical trials (Figure 1) [12,13]. Taxanes bind with β-tubulin with high affinity via π–π interactions [14], and are characterized to promote the polymerization of microtubules and cause the formation of stable, dysfunctional microtubules, thereby preventing normal cellular processes and arresting cell cycle progression [15]. Accumulated data indicate that in addition to antimitotic function, taxanes affect diverse oncogenic signaling pathways, including apoptosis, angiogenesis, inflammatory response, and ROS production [9]. However, despite an initial response, resistance to taxane treatment can develop either at the first treatment or after multiple treatments, which could be one of the major causes responsible for relapses and poor outcomes in patients with HNSCC [16]. Hence, a detailed understanding of the mechanisms of resistance development may help in overcoming chemoresistance to prolonged treatment response.

In this review, we provide an overview of taxane therapy applied in different stages of HNSCC, summarizing the new treatment modalities in recent clinical trials. We describe the mechanisms of taxane resistance in HNSCC and highlight the potential strategies to overcome this resistance.

## 2. Taxane in HNSCC Clinical Application

Clinical studies revealed that taxane addition significantly enhanced therapeutic efficacy compared to the regimen without taxanes in HNSCC treatment [17,18,19,20]. Currently, taxanes combined with chemotherapeutic agents or immune checkpoint inhibitors have been broadly used for locally advanced HNSCC and recurrent or metastatic HNSCC (R/M HNSCC), respectively. Although radiotherapy has been recommended for locally advanced HNSCC [21,22], DTX remains one of the most important agents for induction chemotherapy in the treatment of HNSCC. In addition, beyond taxanes, few effective agents have been suggested for post-ICI treatment; taxanes combined with cetuximab is an important regimen for R/M HNSCC [23]. Despite individual taxane adverse events reported, such as grade 3/4 neutropenia and gastrointestinal disorders [24,25], proper hospitalization care and handling can prolong the use of these drugs in patients.

### 2.1. Locally Advanced HNSCC

Combination chemotherapy with radiotherapy (RT) for patients with locally advanced HNSCC includes concurrent chemoradiotherapy (CRT) and induction chemotherapy (ICT), followed by RT or CRT [26,27]. The most conventional ICT regimen is a combination of cisplatin and 5-fluorouracil (PF) [28]. In order to enhance the effectiveness of ICT, a revolutionary advancement in treatment over the past decades is the integration of docetaxel into the PF ICT regimen. Extensive evidence from two phase III trials confirms that this modified regimen, known as TPF ICT, significantly improves survival rates compared to standard PF ICT [17,18]. Nevertheless, the debate about whether TPF ICT combined with CRT offers superior survival outcomes compared to the standard CRT continues. While two phase III trials demonstrated that TPF ICT combined with CRT did not result in a statistically significant survival advantage over CRT alone [29,30], only one phase III trial of TPF ICT has shown an overall survival benefit compared to no-induction treatment [31]. Notably, a major concern in clinical practice is the occurrence of more severe adverse events, particularly grade 3/4 neutropenia and TPF-related deaths, which can potentially compromise the efficacy of treatment [32,33]. Therefore, the modification of TPF regimens has been developed to lower toxicity and enhance compliance for older patients and the Asian population [8,34]. The TPF regimen is a highly selective treatment for locally advanced HNSCC. Patients who exhibit good partial response and complete response frequently acquire clinical benefits from this treatment [8,35].

### 2.2. R/M HNSCC

For R/M HNSCC, systemic treatments, such as immune check point inhibitor, chemotherapy, and targeted therapy against EGFR, are used as the primary approach to improve long-term disease control; however, their efficacy remains limited [36]. Cetuximab, a monoclonal EGFR antibody, plus cisplatin and docetaxel, have been clinically evaluated for the treatment of R/M HNSCC [37]. The result revealed that progression-free survival was around 6 months in both arms, and most patients developed resistance to treatment within 12 months. Despite less toxicity, this combination therapy was unable to achieve longer survival compared to a conventional Extreme regimen in R/M HNSCC. The advent of immune checkpoint inhibitors targeting programmed death 1 (PD-1) creates a revolutionized HNSCC treatment. Pembrolizumab plus cisplatin and fluorouracil have demonstrated enduring response rates and improved overall survival compared to cetuximab plus platinum chemotherapy [38], although this treatment merely had a median progression-free survival of around 5 months. Additionally, a phase IV trial, the keynote B10 study, demonstrated that pembrolizumab plus paclitaxel and carboplatin had less toxicity and better compliance [39]. In general, taxane backbone treatment plus immune checkpoint inhibitors or cetuximab are better alternatives to R/M HNSCC. In the TPExtreme study, first-line chemotherapy with a taxane (TPEx) followed by a second line with immunotherapy yields a 21.9 month median survival (95% CI 15.9–35.0) [19]. Further study revealed an interesting finding that immunotherapy followed by chemotherapy increased the objective response rate (ORR) to chemotherapy by 30% [20]. Immune checkpoint inhibitors plus taxanes have shown a synergic effect in the phase III KEYNOTE-407 study for squamous non-small-cell lung cancer [40]. A phase I/II study showed that the combination of DTX with pembrolizumab showed promising activity (median OS 21.3 months, 95% CI 6.3–31.1 months) accompanied with a manageable side effect profile in platinum-resistant R/M HNSCC [41].

Collectively, these clinical studies demonstrate an important role of taxanes in chemotherapy for locally advanced HNSCC and R/M HNSCC. The information of recently finished clinical trials in both the area of locally advanced HNSCC and R/M HNSCC and the potential toxicity reported in the trials is summarized in Table 1.

## 3. Molecular Mechanisms of Taxanes

### 3.1. Interfering the Function of Microtubules

The functional mechanism of taxanes was primarily characterized to interfere with the normal functioning of microtubules, which are essential structures involved in cell division and maintenance of cell shape [15]. Taxanes, such as paclitaxel and docetaxel, bind to a beta subunit of tubulin, an essential part of microtubules. This binding stabilizes the microtubules by preventing their disassembly. During cell mitosis, microtubules help separate chromosomes into two daughter cells. Taxanes interfere with the normal formation and function of the mitotic spindle, leading to errors in chromosome segregation and ultimately preventing cell division. By disrupting microtubule dynamics, taxanes induce cell cycle arrest at the G2/M phase. This arrest prevents cells from progressing through the cell cycle and undergoing mitosis, effectively inhibiting cell proliferation and inducing cell death.

### 3.2. Induction of Apoptosis

Taxanes also trigger the signaling pathways of apoptosis. Taxanes can activate several pro-apoptotic signaling pathways within cancer cells. One key pathway is involved in the activation of the c-Jun N-terminal kinase (JNK) and p38 kinase. These kinases phosphorylate and activate c-Jun and ATF-2, which induce the pro-apoptotic signaling [51]. In addition, taxanes disrupt the function of anti-apoptotic proteins, such as Bcl-2 and Bcl-xL, which normally prevent apoptosis by inhibiting the release of cytochrome c from the mitochondria [52]. Taxanes modulate the expression or activity of these anti-apoptotic proteins, promoting the release of cytochrome c and the activation of the caspase cascade. Additionally, taxanes induce the generation of reactive oxygen species (ROS). Taxanes interfere with mitochondrial function, leading to an imbalance in electron transport chain activity and an increase in electron leakage. This electron leakage can result in the formation of superoxide radicals (O_2_^−^) within the mitochondria, which then undergo enzymatic dismutation to form hydrogen peroxide (H_2_O_2_), a type of ROS [53]. Taxanes also stimulate the activity of NADPH oxidase to generate ROS. NADPH oxidase can produce superoxide radicals by transferring electrons from NADPH to molecular oxygen [54,55]. The metabolism of taxanes can result in the formation of reactive metabolites that are involved in inducing oxidative stress and contributing to ROS production. Moreover, taxanes can disrupt calcium signaling within cells. This calcium imbalance is able to activate nitric oxide synthase and xanthine oxidase, thereby producing ROS [56,57]; the accumulation of ROS could finally cause intracellular apoptosis.

### 3.3. DNA Damage and DNA Repair Inhibition

Taxanes have been shown to suppress a set of DNA repair-related genes, whose silence is associated with cancer cell death [58]. Overexpression of DNA repair genes is usually identified in solid tumors harboring chromosomal instability (CIN), a feature characterized as an increase of numerical and structural changes in chromosomes. CIN has been well recognized to be significantly associated with the acquisition of drug resistance in a cell model [59]. The OV01 clinical trial evaluation further indicated that a high level of CIN could contribute to taxane resistance, suggesting CIN may serve as a predictor for response to taxane treatment [58]. A retrospective cohort study showed that the combined use of paclitaxel could benefit the survival of patients with homologous recombination (HR)-deficient ovarian cancers [60]. On the other hand, paclitaxel can inhibit the overexpression of nucleotide excision repair (NER)-related genes induced by doxorubicin, thereby improving the efficacy of the dose-dense ACT protocol for breast cancer treatment [61]. These findings suggest the functional roles of taxanes in modulating DNA repair mechanisms, including nucleotide excision repair (NER) and homologous recombination (HR).

The molecular mechanisms of taxanes in tumor cells are summarized in Figure 2.

## 4. Taxane Resistance in HNSCC

Chemotherapy is a cancer treatment modality that employs drugs to eliminate tumor cells. A typical chemotherapy protocol consists of administering several drugs in cycles of three weeks. However, there are compelling arguments supporting the use of dose-dense protocols in various cancers, including HNSCC. Pharmacodynamics and pharmacokinetics both play crucial roles in the development of taxane resistance (Figure 3). To understand the effect of drug treatment on the organism (pharmacodynamics), it is essential to consider both intrinsic characteristics and the extrinsic tumor microenvironment (TME). There are four primary intrinsic mechanisms associated with taxane resistance: DNA/RNA damage repair, drug efflux, apoptosis inhibition, and abnormal expression of tyrosine kinase pathways. The extrinsic TME involves angiogenesis and interactions between immune cells and tumor cells.

### 4.1. Intrinsic Mechanisms

#### 4.1.1. DNA/RNA Damage Repair

DNA damage and aberrant repair is important in the tumorigenesis, treatment response, and prognosis of HNSCC. A retrospective study of 170 patients with HNSCC reported that among DNA damage response (DDR genes), 17.6% of patients have BRCA2 and ARID1A mutations and 13.5% have ATM mutation, followed by 10% of patients who have BRCA1 mutation [62]. In addition, germline variation in DNA repair genes was substantially identified in patients with young-onset HNSCC, and approximately 67% of these patients had at least one germline variation [63]. The whole-exome sequencing analysis revealed that germline variants in FANCG, CDKN2A, and TPP genes were critical risk factors for HNSCC [63]. Moreover, abnormal expression of DDR-related genes could dramatically impact the treatment response of taxanes in HNSCC. A study involving 453 patients with HNSCC found that the dysregulated expression of ERCC1 and XPA was associated with inferior overall survival and chemotherapy resistance in patients with squamous cell carcinoma of oral cavity [64]. A high expression of DNA repair-related genes (MRE11A, Rad50, RAD51, and XRCC2) in patients with HNSCC also contributes to tumor progression and induces drug-resistant phenotypes [65]. In our laboratory, we found that the combination of arsenic trioxide (ATO) and EGFR inhibitors exhibits a synergistic inhibitory effect on HNSCC. This combination impairs the DNA damage repair response by suppressing the BRCA1-PLK1 signaling pathway. Our single-case experience with a heavily treated patient with HNSCC who had TP53 and BRCA2 mutations showed an unusual and prolonged response to ATO and osimertinib, suggesting the potential to target the DNA damage repair pathway in patients with HNSCC [66].

Due to frequent mutations of DDR-related genes in tumor cells, targeting the DDR pathway has emerged as a novel strategy for cancer treatment. The poly-ADP-ribose polymerases (PARPs) are a group of enzymes catalyzing a mono-ADP-ribose posttranslational modification known as PARylation [67]. PARPs participate in the maintenance of genome integrity as well as playing a critical role in HR DNA repair, especially in tumor cells with mutated BRCA1/BRCA2. It binds to nuclear DNA single-strand breaks (SSBs) and recruits various DNA repair proteins, such as XRCC1, to facilitate DNA damage repair [68]. PARP inhibitors have been reported to provide significant clinical benefits in various solid tumors, including HNSCC [69,70]. Several clinical trials combining PARP inhibitors with taxanes are in progress [71,72,73].

Other than PARPs, the protein kinase Wee1 is another important candidate for targeting DDR-associated pathway, particularly in p53-deficient tumors. Wee1 phosphorylates cyclin-dependent protein kinase 1 (CDK1), a cell cycle checkpoint, to induce G2/M phase arrest, thereby preventing entry into mitosis in response to DNA damage [74]. Therefore, Wee1 suppression abrogates the G2/M checkpoint control that induces and prolongs DNA damage, ultimately leading to mitotic catastrophe and cell death [75]. Moreover, Wee1 inhibitor adavosertib shows the potential for overcoming cisplatin resistance, and the combination of adavosertib and cisplatin inhibits tumor proliferation and survival with synergy in cisplatin-resistant HNSCC by inducing DNA damage [76]. A clinical evaluation revealed that adavosertib in combination with neoadjuvant cisplatin and docetaxel was safe and well-tolerable and showed promising antitumor efficacy in patients with advanced HNSCC, implicating Wee1-targeting as a potential regimen for this disease [73].

#### 4.1.2. Drug Efflux

Accumulating evidence indicates that the adenosine triphosphate-binding cassette (ABC) transporter actively pumping chemotherapeutic drugs out of cells is the main cause of tumor multidrug resistance (MDR) [77]. the overexpression of ABC transporters, including P-glycoprotein (P-gp), breast cancer resistance protein (BCRP), and multidrug resistance-associated protein (MRP), has been implicated in a serious therapeutic obstacle [78]. For instance, high levels of P-gp are dramatically associated with taxane resistance in many experimental settings [79]. Therefore, the inhibition of P-gp activity has been explored as a potential approach. For instance, pentagalloyl glucose was investigated for its inhibitory effects on P-gp and its impact on cancer stem-like cells in HNSCC [80]. Additionally, cabazitaxel is a derivative of docetaxel, which is cytotoxic to docetaxel-resistant cell lines due to P-gp overexpression. Cabazitaxel has been shown to improve the survival of prostate cancer patients who experienced relapse after docetaxel treatment [81]. A phase II clinical trial was conducted to evaluate the efficacy of cabazitaxel compared to docetaxel in recurrent head and neck cancer patients. Unfortunately, cabazitaxel did not show superior disease control compared to docetaxel [82]. Further investigation is needed to determine whether cabazitaxel can provide benefits to patients who have progressed after docetaxel treatment. One of the challenges concerning the use of cabazitaxel is the high frequency of grade 3 neutropenia and febrile neutropenia, which critically precludes its combined use with other cytotoxic agents in HNSCC [83]. Other novel taxanes, such as liporaxel and tesetaxel, also demonstrated poor affinity to P-gp and better antitumor efficacy. However, major side effects, such as neuropathy and neutropenia, and relatively limited clinical efficacy have postponed further clinical development [13,84]. In this regard, the application of drug-conjugated antibodies may point to a new direction in using cytotoxic agents against cancer cells with low systemic toxicities. Recently, monotherapy with enfortumab vedotin, a lectin-4-directed antibody conjugated with a microtubule inhibitor (MMAE), obtained an overall response rate of 24%, as well as good tolerability in patients with heavily treated head and neck cancer in a phase II EV-202 study [85]. Until now, the paclitaxel-based drug-conjugated antibodies have not advanced into clinical trials since paclitaxel and its analogs display insufficient cytotoxicity in many preclinical evaluations [13]. However, several studies revealed that introducing a hydrophilic linker to the ultra-hydrophobic paclitaxel can markedly enhance antitumor activities [86,87]. Although much of the literature focuses on the discussion of the overexpression and functions of P-gp, diverse ABC transporters have been linked to the resistance mechanism. In animal models, Abcc10^+/+^ tumors are associated with decreased apoptosis and metastasis, while Abcc10^−/−^ lines increase sensitivity to DTX and PTX [88]. Therefore, the compensatory roles of multiple ABC transporters should be under consideration during pursuit of the related targeting therapies.

#### 4.1.3. Apoptosis Inhibition

Apoptosis inhibition has been observed in taxane-resistant HNSCC. Liu et al. indicated that HNSCC cells can upregulate survivin expression by activating the NF-κB pathway. Survivin inhibits active caspase-9, thereby preventing apoptosis induced by aberrant mitosis resulting from mitotic damage caused by paclitaxel [89]. Other inhibitors of apoptosis proteins (IAP), including XIAP (X-linked inhibitor of apoptosis), cIAP1/2 (cellular inhibitor of apoptosis 1 and 2), and SMAC (second mitochondria-derived activator of caspases), have also been found to play crucial roles as survival factors in HNSCC [90]. Currently, clinical trials are underway to investigate the use of IAP antagonists for targeting HNSCC. An exploratory trial is designed to study patients with HNSCC who received xevinapant, cisplatin chemotherapy, or both treatments before surgery [69]. After xevinapant treatment, tumors exhibited higher infiltration by CD8+ T cells and NF-κB/IAP pathway mediators [91]. Further investigation is needed to explore the combined effects of taxanes and apoptosis inhibitors. Although the inhibition of apoptosis has shown promising results in overcoming taxane resistance, the effects of these inhibitors depend on the cells’ physiological state and gene expression status. In addition, the cell death pathway of cancer cells may be dynamic and can escape through its dependence on apoptosis, ferroptosis, necroptosis, or even senescence [92]. Therefore, the profiling of apoptosis regulators might be useful in identifying the best drug combinations [93].

### 4.2. Extrinsic Mechanisms

#### 4.2.1. Angiogenesis

Angiogenesis involved in nutrient supplements contributes to tumor proliferation and metastasis. Accompanying tumor growth, the formation of abnormal blood vessels and lymphatic vessels leads to the accumulation of interstitial fluid in the interstitium. This phenomenon elevates tumor interstitial pressure, which generates a physical barrier to the infiltration of immune cells and chemotherapeutic drugs [94]. Angiogenesis is a hallmark of tumor progression, and targeting it has proven to be a successful approach in treating certain solid tumors. Bevacizumab, in combination with chemotherapy, has received approval from the US FDA for the treatment of several malignancies [95]. It has been demonstrated that the activity of taxanes is partially dependent on their impact on capillary tube formation and microvessel density [9]. The impact of taxanes on angiopoietin-1 (Ang-1) and VEGF-A signaling pathways, which are relevant for the maintenance and proliferation of endothelial cells, has been described in several solid tumors [96,97]. Therefore, it is rational to combine taxanes with angiogenesis inhibitors to enhance antitumor activity. Because preclinical data showed promising results for bevacizumab in the treatment of HNSCC [98,99], numerous clinical trials investigated bevacizumab in combination with chemotherapy [100,101,102]. Although these studies showed promising response rates, concerns about toxicities such as perforation and hemorrhage persisted. A large phase III trial was conducted to assess the addition of bevacizumab to chemotherapy regimens. This trial revealed a significant improvement in progression-free survival and the overall response rate. However, it did not demonstrate a statistically significant survival advantage. Notably, bevacizumab addition significantly increased the risk of high-grade bleeding (grade 3–5; 6.7% vs. 0.5%; *p* < 0.001) and treatment-related mortality (9.3% vs. 3.5%; *p* = 0.022) [100]. Regarding tyrosine kinase inhibitors of angiogenesis, compounds such as sorafenib, sunitinib, semaxanib, levantinib, and vandetanib have been studied to enhance the response rate to chemotherapy [103,104,105]. However, the combination of tyrosine kinase inhibitors with chemotherapy, including docetaxel and paclitaxel, showed only limited clinical significance.

#### 4.2.2. The Interaction of Immune Therapy and Chemotherapy

The feature of immune-escape protecting tumor cells from immune attacks is commonly characterized in various types of cancers, including HNSCC. The programmed cell death protein 1 (PD-1) homologous to CD28 functions in the promotion of self-tolerance, regulation of adaptive immune responses, and inhibition of immune signaling. The interaction between PD-1 and its programmed cell death ligand-1 (PD-L1) is the most representative T cell immune checkpoint, and it is broadly applied for drug development [106]. Targeting T cells with PD-1/PD-L1 inhibitors in combination with chemotherapeutic drugs has been the standard of care in metastatic/recurrent HNSCC. It is still unclear whether the combination effects of PD-1/PD-L1 inhibitors and chemotherapies are synergistic or additive. Several retrospective studies have shown that exposure to immune checkpoint inhibitors improves the response to salvage chemotherapy in HNSCC, suggesting that immune checkpoint inhibitors may increase tumor sensitivity to chemotherapy [20,107].

Other than T cells, other immune cells, such as tumor-associated macrophages (TAMs), also play important roles in tumorigenesis and chemotherapeutic resistance in HNSCC. In addition to microtubule stabilization, PTX could activate the TLR4 receptor and reprogram M2-polarized macrophages to the M1-like phenotype, thereby inducing a proinflammatory response to increase antitumor [108]. In our laboratory, we found that tumor-associated macrophages are associated with DTX resistance [54]. Through increased IL-1β secretion, macrophage coculture enhances the expression of intercellular adhesion molecule 1 (ICAM1), which promotes the stemness property and the formation of polypoid giant cancer cells to increase the resistance of HNSCC to DTX. Interestingly, ATO treatment decreases macrophage infiltration into the TME and attenuates IL-1β secretion using macrophages in the preclinical analysis. Moreover, the combination of ATO and DTX synergistically suppresses tumor growth in a mouse model [54]. Although targeting IL-1β or ICAM1 is a promising concept, there are no clinical trials that have assessed targeting IL-1β or ICAM1 for HNSCC treatment, and more research in this area is warranted.

#### 4.2.3. Optimizing the Pharmacokinetics of Chemotherapy

Dose-dense chemotherapy increases the dose intensity of the regimen by delivering standard-dose chemotherapy with shorter intervals to avoid the regrowth of cancer cells. Dose-dense chemotherapy can help overcome chemotherapy resistance in some cases. By delivering chemotherapy more frequently and at higher doses, it may prevent cancer cells from developing resistance mechanisms and reduce the chances of tumor growth. This approach is particularly relevant in the treatment of certain aggressive cancers, such as breast cancer, bladder cancer, and lymphoma [109,110]. However, the associated studies in HNSCC are scarce. Our team conducted a phase II clinical trial using biweekly TPF rather than the conventional triweekly TPF regimen. The data showed that this biweekly TPF induction chemotherapy regimen had an overall response rate of 89.7% and a complete response rate of 31%. Notably, the grade 3–4 neutropenia rate was lower than that of the conventional triweekly TPF regimen [8]. A retrospective study by J. Fayette et al. in France reported a similar biweekly regimen for metastatic head and neck HNSCC and also confirmed a high response rate with tolerable toxicity [111]. A randomized phase II prospective study conducted in India showed that the biweekly regimen had better response rates with fewer toxicities compared to conventional triweekly regimens [112]. In summary, dose-dense regimens are a promising approach in patients with HNSCC. Further investigation is warranted to determine the appropriateness of this concept in combination with targeted therapies or immune therapies.

## 5. Ongoing Clinical Trials

Accumulating evidence from clinical trials has revealed that taxanes are among the most active antitumor agents currently available for HNSCC. However, the use of taxane monotherapy has rarely been recommended for the first-line treatment of HNSCC [113]. Recently, several studies showed partial or complete responses to taxane monotherapy after ICI in R/M HNSCC [114,115,116], suggesting the potential benefit of taxane monotherapy in third-line treatment [113]. Instead of use alone, PTX and DTX usually combine with other regimens, such as chemotherapeutic drugs, radiation, and ICIs, to reach better efficacy in locally advanced and R/M HNSCC. Currently, many clinical trials are ongoing for finding the optimal drug combination and treatment condition, and the information is summarized in Table 2.

## 6. Conclusions

Currently, taxanes, comprising PTX and its synthetic derivatives, have been broadly used for chemotherapy in diverse cancers, including HNSCC. In addition to stabilizing β-tubulin polymerization to suppress microtubule dynamics and impair mitosis, the functions of taxanes were characterized to be involved in many oncogenic signaling pathways, such as angiogenesis, inflammatory response, ROS production, and apoptosis induction. However, the intrinsic and acquired resistance remains a major obstacle to prolonged treatment response. A detailed understanding of the mechanisms of resistance may lead to the development of dual-targeting compounds and effective combination strategies for surmounting drug resistance.

## Figures and Tables

**Figure 1 biomedicines-11-02887-f001:**
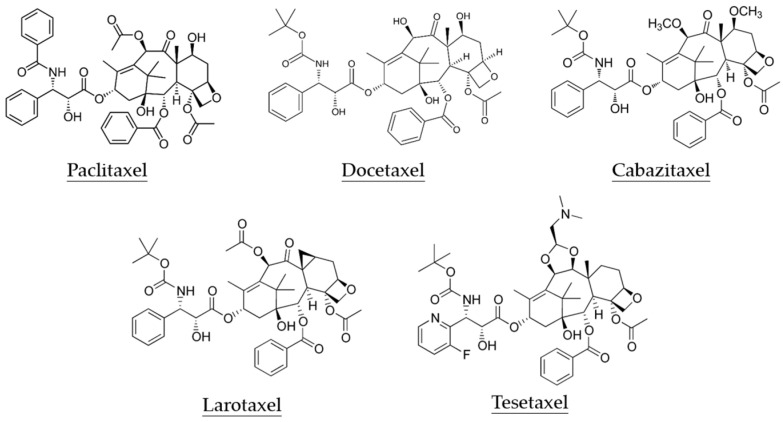
Chemical structures of taxanes used in clinical settings (**upper panel**) and oral taxanes currently under development (**lower panel**).

**Figure 2 biomedicines-11-02887-f002:**
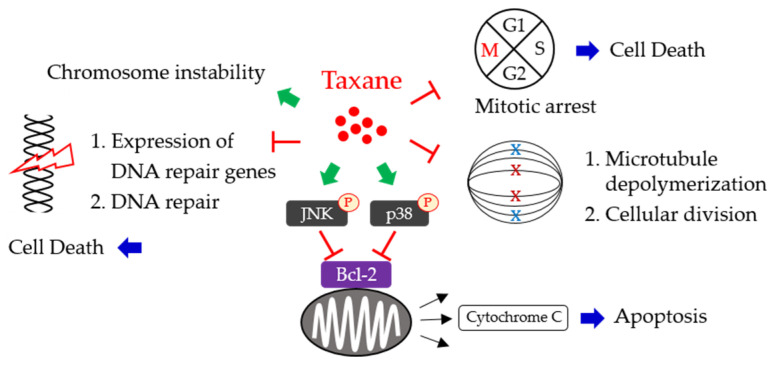
Schematic representation of the major mechanisms of taxanes involved in inducing tumor cell death.

**Figure 3 biomedicines-11-02887-f003:**
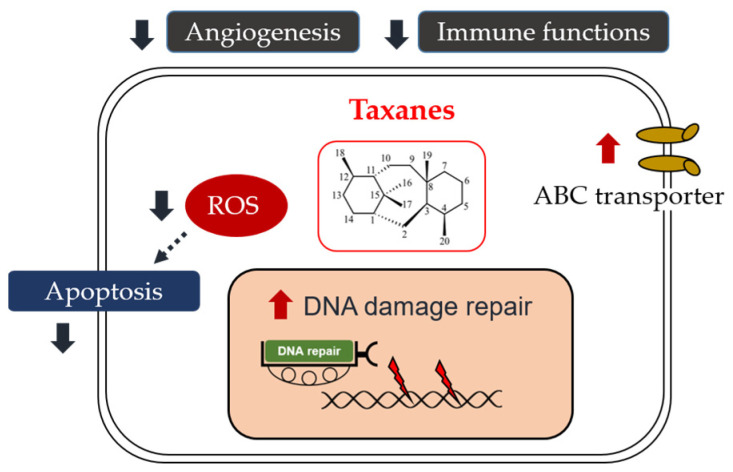
Resistant mechanisms of taxanes in HNSCC.

**Table 1 biomedicines-11-02887-t001:** Recently finished trials of taxanes in locally advanced HNSCC and R/M HNSCC.

A. Locally Advanced HNSCC								
				Patient Number	Major Adverse Event (AE)			
	Trial Design	Arms	Combined Agents		AE	N (%)	Grade	Reference
	Open-label, Single-center	1	Toripalimab + Paclitaxel/Cisplatin	20	Baldness	20 (100)	1–2	[42]
	Randomized, Open-label, Multi-center	2	RT ± Docetaxel	356	Odynophagia	177 (98.9)	3–5	[43]
	Randomized, Open-label, Multi-center	2	TPF, cisplatin + RT ± Cetuximab + RT	519	Mucositis	161 (79.7)	2–3	[44]
	Randomized, Single-center	2	Cisplatin/Carboplatin, Docetaxel + Erlotinib vs. placebo	100	Fatigue	40 (77)	1–2	[45]
	Open-label, Multi-center	1	Panitumumab (Pb) + Paclitaxel, followed Bio-RT + Pb	51	Skin toxicity	29 (56.9)	3–4	[46]
	Randomized, Open-label, Multi-center	2	TPF or TPC, followed RT + Cetuximab	100	GI disorders	23 (23)	3–5	[47]
	Non-randomized, Single-center	2	Nanoparticle albumin-bound Paclitaxel, RT/Cisplatin vs. RT/Cetuximab	79	Fatigue	44 (55.7)	1–2	[48]
**B. R/M HNSCC**								
				**Patient Number**	**Major Adverse Event (AE)**			
	**Trial Design**	**Arms**	**Combined Agents**		**AE**	**N (%)**	**Grade**	**Reference**
	Open-label, Multi-center	1	Pembrolizumab + Paclitaxel + carboplatin	92	Neutropenia	52 (57)	1–3	[39]
	Open-label, Multi-center	1	durvalumab combined with weekly paclitaxel carboplatin	64	no disclosure			[49]
	Open-label, Single-center	1	Pembrolizumab plus docetaxel	22	Neutropenia	3 (13.6)	3–5	[41]
	Open-label, Multi-center	1	Nab-paclitaxel+ cetuximab+carboplatin	74	Neutropenia	25 (34)	3–5	[50]

**Table 2 biomedicines-11-02887-t002:** Currently ongoing clinical trials of PTX and DTX in HNSCC.

Agent	NCT Identifier	Phase	Study Design	Population
Paclitaxel				
	NCT04338399	III	Buparlisib and Paclitaxel vs. Paclitaxel	R/M HNSCC post PD-1 or PD-L1 inhibitor
	NCT05420948	II	1.Pembrolizumab alone	Circulating tumor DNA Response-Adaptive Pulsed Chemotherapy in R/M HNSCC
			2.Pembrolizumab+Paclitaxel+Carboplatin	
	NCT02270814	II	Nab-Paclitaxel, Platinum, Cetuximab	Incurable HNSCC (R/M + unresectable HNSCC)
	NCT04282109	II	Nivolumab/Paclitaxel	Cisplatin refractory R/M HNSCC
	NCT04278092	II	Cetuximab + Paclitaxel	R/M HNSCC post first line Pembrolizumab
	NCT04831320	II	Nivolumab/Bab-Paclitaxel	R/M HNSCC post PD-1 or PD-L1 inhibitor
	NCT03440437	I/II	FS118 + Paclitaxel (expansion cohort)	R/M HNSCC post PD-1 or PD-L1 inhibitor
	NCT05283226	II	Oral NRC-2694-A + Paclitaxel	R/M HNSCC post PD-1 or PD-L1 inhibitor
	NCT04858269	II	Pembrolizumab + weekly CT (Paclitaxel + Carboplatin)	R/M HNSCC first line
	NCT05758389	II	Tislelizumab, Paclitaxel (albumin-bound type), Cisplatin, 5-FU	Neoadjuvant CT for newly diagnosed resectable advanced HNSCC
	NCT04826679	II	Camrelizumab, Nab-Paclitaxel, Cisplatin	Neoadjuvant CT for newly diagnosed resectable advanced HNSCC
	NCT05459129	I/II	Atezolizumab + Tiragolumab + Carboplatin + Paclitaxel	Neoadjuvant CT for newly diagnosed resectable advanced HNSCC
Docetaxel				
	NCT05057247	II	Duvelisib Plus Docetaxel	R/M HNSCC post PD-1 inhibitors
	NCT05252429	II	Pembrolizumab/Docetaxel	R/M HNSCC first line
	NCT05376553	II	Cemiplimab + Docetaxel/Cisplatin	Induction setting for LAHNSCC
	NCT04722523	II	Cemiplimab/Cetuximab/Docetaxel/platinum	Resectable LAHNSCC Neoadjuvant
	NCT05726370	II	Pembrolizumab/Platinum/Docetaxel	Resectable R/M HNSCC first line Neoadjuvant

## Data Availability

Not applicable.

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
