# Peer review of "Taxanes in the Treatment of Head and Neck Squamous Cell Carcinoma"

_biomedicines, 2023, doi:10.3390/biomedicines11112887_

Round 1
Reviewer 1 Report
This MS presents an overview of taxanes’ therapy used in various stages of HNSCC, depicts the mechanisms of taxanes’ resistance, and describes some effective combination approaches for surmounting drug resistance in HNSCC.
The Authors can consider some suggestions for the revision.
In the whole MS, please, use the phrase: ‘patients with HNSCC’ rather than: ’HNSCC patients’.
In the Introduction [p 2], in the sentence:
‘Accumulating data indicate that except antimitotic function, taxanes affect diverse oncogenic signaling pathways including apoptosis, angiogenesis, inflammatory response, and ROS production [9].’
The word: ‘except’ should be replaced by: ‘in addition to’.
‘Accumulating data indicate that in addition to antimitotic function, taxanes affect diverse oncogenic signaling pathways including apoptosis, angiogenesis, inflammatory response, and ROS production [9].’
A table will be helpful to concisely summarize the recent RCTs’ results in the area of locally advanced HNSCC (including concurrent CRT or ICT, followed by RT or CRT), and R/M HNSCC (including, the systemic treatments, like ICIs , chemotherapy, and targeted therapy against EGFR). The potential toxicity of these agents should be underlined.
In the Conclusion, please modify the last two sentences, as suggested below [from 1 to 2]:
1.In addition to stabilize β-tubulin polymerization to suppress microtubule dynamics and impairs mitosis, the functions of taxanes were characterized to be involved in many oncogenic signaling pathways such as angiogenesis, inflammatory response, ROS production, and apoptosis induction.
2.In addition to stabilizing β-tubulin polymerization to suppress microtubule dynamics and impair mitosis, the functions of taxanes were characterized to be involved in many oncogenic signaling pathways such as angiogenesis, inflammatory response, ROS production, and apoptosis induction.
1.However, the intrinsic and acquired resistance remains a major obstacle to prolonged treatment response. Through a detailed understanding the mechanisms of resistance may lead to emerge the development of dual targeting compounds and the effective combination strategies for surmounting drug resistance.
2.However, the intrinsic and acquired resistance remains a major obstacle to prolonged treatment response. A detailed understanding of the mechanisms of resistance may lead to the development of dual-targeting compounds and effective combination strategies for surmounting drug resistance.
A list of abbreviations needs to be added at the end.
Minor editing of the English language required
Reviewer 2 Report
biomedicines-2650311, Taxanes in the treatment of head and neck squamous cell carcinoma, by Ching-Yun Hsieh and al.
The manuscript is a targeted review that provides a general look on the use of taxanes as treatment for head and neck squamous cell carcinoma. Overall the quality of the paper is acceptable, but it could be greatly improved.
The authors should present the chemical structures of paclitaxel and all others taxanes that have been clinically tested for cancer. The authors seems to focus only on paclitaxel, docetaxel, and cabazitaxel. The article should present all the taxanes in clinical studies, like larotaxel or tesetaxel, to name just a few.
In section 2 of the paper the authors need to introduce a section to describe the particularities of the head and neck squamous cell carcinomas and to justify why the taxanes are a good choice for the treatment. They should add also a section to briefly describe the other solutions beyond taxanes and argue the advantages or disadvantages of taxanes.
The authors should prepare and present in section 3 a figure to highlight the mechanism of action for the taxanes.
There should be a new section, before the conclusions, to present the clinical studies performed or underway focused on taxanes as single agents or combinations as treatment for HNSCC.
The authors should critically review their paper considering the information it conveys. It should be more than a simple collection of data, most of it known by the readers working in this field. It should provide a solid base of information with a strong input from the authors.
There are many review work on taxanes, that in my opinion provide a better look at this class of compounds. I advise the authors to study them carefully in order to improve their work and most importantly to highlight clearly for the readers what this new work adds new.
Here are some examples:
Novel taxanes in development: Hopes or hypes? Critical Reviews in Oncology/Hematology, 176, August 2022, 103727
A systemic review of taxanes and their side effects in metastatic breast cancer, Front Oncol. 2022; 12: 940239
The Evolving and Future Role of Taxanes in Squamous Cell Carcinomas of the Head and Neck
The Evolving Role of Taxanes in Combination With Cetuximab for the Treatment of Recurrent and/or Metastatic Squamous Cell Carcinoma of the Head and Neck: Evidence, Advantages, and Future Directions
The references cited by the authors a very limited and I advise them to expand their number.
OK
Round 2
Reviewer 2 Report
The authors changed the manuscript according to the comments of the review improving the quality of their work. They should remove the structure they called "oraxol"' from figure 1. The correct name is encequidar, and is NOT a taxane, but a P-glycoprotein inhibitor. They should be careful in the description of oraxol that is fact is a mixture, not a single molecules. Similarry, liporaxel is a commercial name that should not be used and it represents a special oral formulation for paclitaxel. It is NOT a new taxane. The authors should find a pharmacist or a pharmacology expert to correct other such mistakes that maybe I did not find.
OK
